# Prevalence and determinants of anemia among HIV-positive children in central Uganda: A cross sectional study

Abdirashid Dahir Herow **⬤***, Ndeezi Grace, Joel Kimera, Martin Nduwimana

Department of Paediatrics and child health, Faculty of clinical medicine and dentistry Kampala International University-Western Campus, Bushenyi, Uganda

* xeroow1818@gmail.com

## Abstract

### Background

Anemia remains a critical public health issue, especially in low-resource settings like Uganda, with severe consequences especially in HIV-positive children. Limited research has been conducted in low-income settings especially about anaemia in HIV-Positive children. This study aimed to determine the prevalence and factors associated with anemia among HIV positive children at Kayunga Regional Referral Hospital. In addition to assess iron deficiency anemia using Mentzer index.

### Methods

This was a hospital based cross-sectional study conducted at KRRH. 384 HIV-positive children aged 6 months to 12 years attended the KRRH ART clinic between November 2024 to January 2025 were enrolled.Data was collected using structured questionnaires,information regarding their sociodemographic, medical and laboratory was obtained asking care giver or reviewing the medical record. Anthropometric measurements were done and interpreted compared to the WHO z-scores and obtained blood sample for CBC and analyzed.
Anemia was defined based on age-specific hemoglobin thresholds: < 11.0 g/dL for children aged 6–59 months, < 11.5 g/dL for those aged 5–11 years, and <12.0 g/dL for children aged 12–14 years. Iron deficiency anemia was diagnosed with MI > 13. Logistic regression in STATA was done to determine the factors that were significantly associated with anaemia.

### Results

Majority of the children (63.5%) were over 5 years of age and 52.9% were male and 57.3% had anemia with 95% confidence interval of 52.3–62.2%. The predominant severity category of anemia was moderate (48.6%). Multivariable analysis showed

**Data availability statement:** All relevant data are within the paper and its Supporting Information files.

**Funding:** The author(s) received no specific funding for this work.

**Competing interests:** The authors have declared that no competing interests exist.

**Abbreviations:** aOR: Adjusted odds Ratio; ART: anti-retroviral treatment; AIDS: Acquired immune deficiency syndrome; CD4: Cluster of differentiation; cOR: Crude Odds ratio; HIV: Human immune deficiency virus; KRRH: Kayunga Regional Referral Hospital; LMICs: Low and middle-income countries;MUAC: Mid-upper arm circumference; stata; WHO; WAZ-Weight-for-age Z score; WHZ: Weight-for-height Z score; WLZ: Weight-for length Z score.

that caretakers who stopped in primary, having more than four siblings, being in current HIV stages 2 or 3, any history of hospital admission in the preceding six months, and being severely stunted, wasted, or experiencing concurrent stunting and wasting were independently associated with anemia among children infected with HIV (P-value < 0.05 for all). Iron deficiency anemia accounted for 54.1% of all children with anemia.

## Introduction

The World Health Organization (WHO) defines anemia in children based on hemoglobin levels, with thresholds varying by age and sex [1]. Human Immunodeficiency Virus (HIV) exacerbates anemia through multiple mechanisms, including chronic inflammation, opportunistic infections, and nutritional deficiencies [2,3].

Globally, anemia affects approximately 42% of children under five years, with the highest prevalence in sub-Saharan Africa [4]. In Africa, anemia rates among children on antiretroviral therapy (ART) are alarmingly high, with studies reporting prevalence as high as 54.2% [5].

In Uganda, anemia prevalence among HIV-positive children ranges between 50% and 60%, driven by factors such as poverty, inadequate healthcare access, and high HIV burden [6,7].

According to a cross-sectional study conducted on 265 HIV-positive children attending the ART clinic at Gondar University Hospital in Ethiopia between February and June 2013, eating green leafy vegetables and taking cotrimoxazole were linked to anemia. However, no significant correlation was seen with the percentage of cluster of differentiation 4 (CD4) cells, opportunistic infections, sex, age, or WHO clinical stage [8].Being severely immunosuppressed, having undernutrition, taking zidovudine-based antiretroviral therapy, and having tuberculosis were all found to be independent predictors of anemia in children receiving ART among 391 children in Ethiopia [9].

Globally, studies have shown that HIV-related anaemia is highly prevalent, yet its screening, diagnosis, and management remain suboptimal in many settings [10]. In Uganda, national anaemia policy of 2002 recommend routine screening for anaemia in all HIV-positive children and provision of preventive treatment [11]. However, there is limited evidence on the extent to which this guideline is implemented, particularly in regional referral hospitals such as Kayunga Regional Referral Hospital (RRH).

Despite global and regional efforts to combat anemia, most studies on the HIV-anemia interplay have focused on adults in developed nations, with limited research specifically addressing pediatric populations in low-income settings. In Uganda, while two studies have documented the prevalence of anemia in HIV-positive children; one conducted at Mulago and the other in Mbarara, Western Uganda [6,7], no such study has been conducted in Kayunga. Additionally, these two studies conducted in Uganda did not look at the burden of iron deficiency anemia, which is the most common form of nutritional anemia in this population.There

remains a critical gap in understanding this burden within semi-rural and peri-urban contexts which present distinct socio-demographic and ecological characteristicsincluding lower caregiver education levels, higher household sizesthat may influence nutritional status and anemia risk differently. This study addresses that gap by providing context-specific data on anemia prevalence and its associated factors among HIV-positive children, thereby informing tailored interventions and resource allocation for similar underserved regions.

## Materials and methods

### Study design and setting

This hospital based descriptive cross-sectional study was conducted at KRRH. It is located in Kayunga District, which is in central Uganda, about 67.5 km east of Mulago National Referral Hospital.

Kayunga Regional Referral Hospital services to residents from Kayunga district and neighboring districts including Kamuli, Buikwe, Mukono, Luweero and Nakasongola.

The district is relatively close to Lake Kyoga, a major water body in Uganda. The proximity of Kayunga to Lake Kyoga and its associated ecosystems influences the economic activities of the region, which in turn impact maternal and child nutrition.

This data was collected between 4th November, 2024 and 31st January, 2025 and we included all HIV-positive children aged 6 months to 12 years attendedART clinic during the study period whose caretaker's consented to participate. We excluded those who had other chronic conditions like leukemia and sickle cell disease. In addition, severely ill children were excluded to avoid delaying treatment but could be included later when stable.

### Sampling and justification

Sample size calculation formula by KishLesliewas used to determine the Sample size for this study, using the estimated prevalence of 50.7% based a study conducted by(6)and the calculated sample size was384 participants.

The prevalence of 50.7% was selected to maximize statistical efficiency, as it yields the largest sample size for a given precision. With a 95% confidence level and a 5% margin of error, the calculated sample size ensured adequate power to detect associations between anemia and key sociodemographic and clinical factors. Consecutive sampling was employed until the target sample size was reached.

### Data collection tools

Structured questionnaires were used to collect sociodemographic and clinical data from caregivers and medical records. Prior to data collection, the questionnaires were pilot-tested on a sample of 20 HIV-positive children who visited Nkokon-jeru health facility. This pilot exercise helped refine question clarity, flow, and cultural appropriateness. Feedback from the pilot was used to revise ambiguous items and ensure consistency in data capture. The final version was administered by trained research assistants fluent in both English and Luganda.

### Data collection procedures

Information on sociodemographic characteristicsincluding age, sex, residence, birth order, household size, caretaker's age, relationship to the child, education level, occupation, and family monthly incomewas collected using a structured questionnaire. These data were obtained from caregivers and cross-verified with the participants' medical records.

Clinical data were extracted from medical records and included viral load, CD4 count, HIV clinical staging, duration since HIV diagnosis, number of hospital admissions in the preceding six months, history of fever, ART regimen, history of opportunistic infections (e.g., oral candidiasis, tuberculosis), history of malaria, blood transfusion, and use of nutritional supplements such as iron and multivitamins. Viral load was categorized as <200, 200–500, and >500 copies/mm³, while

CD4 count was classified as detected or undetected. HIV clinical staging was assessed during the study period by the principal investigator and trained research assistants.

Nutritional status was assessed using standard anthropometric measurements, including weight-for-height (WHZ), height-for-age (HAZ), and weight-for-age (WAZ) Z-scores, interpreted according to WHO Child Growth Standards. Stunting, wasting, and underweight were classified as follows:

**Moderate stunting/wasting/underweight:** Z-score between –2 and –3 standard deviations (SD)

**Severe stunting/wasting/underweight:** Z-score less than –3 SD

Children with concurrent stunting and wasting (WaSt) were identified when both HAZ and WHZ were below –2 SD. These classifications enabled consistent identification of undernutrition severity and its association with anemia.

For laboratory analysis, 2 ml of venous blood was collected from each participant after obtaining informed consent from the caregiver and assent from the child. The venipuncture site was disinfected with an alcohol swab, and blood was drawn into EDTA-containing vacutainers to prevent clotting. Complete blood count (CBC) was performed using an automated Coulter counter, which measured hemoglobin concentration, red blood cell indices (MCV, MCH, MCHC), packed cell volume, red cell distribution width, white blood cell count, and platelet count.

Red blood cell morphology was classified as:

• Normocytosis: MCV = 76–96 fL

• Microcytosis: MCV < 76 fL

• Macrocytosis: MCV > 96 fL

• Normochromic: MCHC = 31–35 g/dL

• Hypochromic: MCHC < 31 g/dL

Anemia was diagnosed based on WHO age-specific hemoglobin thresholds: < 11 g/dL for children aged 6–59 months, < 11.5 g/dL for ages 5–11 years, and <12 g/dL for ages 12–14 years. Severity was categorized as:

• **Severe:** <7 g/dL

• **Moderate:** 7–8.9 g/dL

• **Mild:** 9 g/dL to the lower limit of normal for age

### Data analysis

This was conducted using STATA software and was structured to address the study objectives.

To determine the prevalence of anemia, the number of participants diagnosed with anemia based on age-specific hemoglobin thresholdswas divided by the total number of enrolled participants. The results were expressed as frequencies and percentages, presented both in narrative form and summarized in tabular format.To identify factors associated with anemia, binary logistic regression analysis was performed. Initially, bivariable analysis was used to explore associations between anemia and selected sociodemographic, clinical, and nutritional variables. Variables with a p-value ≤ 0.20 were considered for inclusion in the multivariable logistic regression model. In the multivariable analysis, statistical significance was set at p ≤ 0.05, and adjusted odds ratios (aOR) with 95% confidence intervals were reported to identify independent predictors of anemia{Hosmer Jr, 2013 #23}.To assess the adequacy of the final model, the Hosmer-Lemeshow goodness-of-fit test was performed to evaluate whether the observed event rates matched expected event rates across deciles of predicted risk. A non-significant p-value (p > 0.05) indicated that the model fit the data well, suggesting that the logistic regression assumptions were met and the model was appropriate for identifying independent predictors of anemia.

Confounding variables such as dietary intake, micronutrient supplementation and malaria status were adjusted for in the final model.

Iron deficiency anemia was assessed using the Mentzer index (MI), calculated as the ratio of mean corpuscular volume (MCV) to red blood cell count (RBC). Children with a MI greater than 13 were classified as having iron deficiency anemia. The proportion of affected children was visually summarized using a pie chart to illustrate the burden of iron deficiency within the anemic subgroup.

### Ethical considerations

Ethical approval for the study was obtained from the Research and Ethics Committee of Kampala International University (KIU-2024–517). Privacy and confidentiality were ensured by individually assessing participants, anonymizing question-naires with number codes, and securely storing data. Written informed consent was obtained from the caretakers/guard-ians of eligible participants aged 7 years and below, while participants aged 8 years and above provided assent by signing an assent-form.

## Results

### Sociodemographic characteristics of study participants

In this study that enrolled 384 HIV-positive children attended antiretroviral treatment clinic at Kayunga Regional Referral Hospital (KRRH), majority were over 5 years of age 244 (63.5%). Slightly over half were male 203 (52.9%). Only 232 (60.4%) of the participants were cared for by their mother. Over half of the families had a monthly income of less than 200,000 Uganda shillings 198 (51.6%) (Table 1).

### Clinical and laboratory characteristics of study participants

In this study that enrolled 384 HIV positive children attended antiretroviral treatment clinic at Kayunga Regional Referral Hospital (KRRH), Majority of the participants had an detected viral load 242 (63.0%), though majority were classified as HIV stage one 299 (77.9%). Majority had been diagnosed for over one year 264 (68.8%) (Table 2).

### Prevalence of anemia among HIV positive children at KRRH

The analysis revealed that 220 of the 384 HIV positive children were found to have anemia, hence a prevalence of 57.3% with 95% confidence interval of 52.3–62.2%. The predominant severity category was moderate, accounting for 107 (48.6%) of all children with anemia. Majority of the anemic children had Microcytosis 181 (82.3%) and Hypochromia 196 (89.1%) (Table 3).

The bivariate analysis results indicated that six socio-demographic factors were significantly associated with anemia (P ≤ 0.2). These factors included age of the child37–60 (cOR =0.237, 95% CI =(0.098–0.571) P value = 0.001, residenceR-ural (cOR=1.846, 95% CI =(1.226–2.779) P value = 0.003, relationship of the care giver(cOR =0.207, 95% CI =(0.089–0.482) P value =**<0.001**, education level of the care taker (cOR =1.950, 95% CI =(1.055–3.605) P value = **0.033**, family monthly income<200,000 (cOR =2.656, 95% CI =(1.251–5.638) P value = **0.011**, number of siblings>4 (cOR =7.876, 95% CI =(2.716–22.844) P value =**<0.001** (Table 4).

Clinical and laboratory factors significantly associated with anemia among HIV positive children includedHIV clinical stage**2** (cOR =2.161, 95% CI =(1.209–3.862) P value = **0.009 ≥ 3** (cOR =3.570, 95% CI =(1.166–10.929) P value = **0.026**, number of admissions in the previous 6 months**1** (cOR =1.594, 95% CI =(0.923–2.753) P value = **0.095 2** (cOR =6.439, 95% CI =(3.269–12.682) P value =**<0.001 3 (**cOR =5.429, 95% CI =(2.279–12.930) P value =**<0.001**, ART regimenAZ-T/3TC/DTG **(**cOR =5.577, 95% CI =(0.678–45.844) P value = **0.110**, white blood cell countLow **(**cOR =1.883, 95% CI =(0.887–3.997) P value = **0.099** High **(**cOR =4.053, 95% CI =(1.824–9.004) P value = **0.001**, platelet countHigh **(cOR

**Table 1. Sociodemographic characteristics of study participants.**

| Characteristic | Frequency | Percentage |
|---|---|---|
| **Age of the child (months)** | | |
| 6-24 | 45 | 11.7 |
| 25-36 | 46 | 12.0 |
| 37-60 | 49 | 12.8 |
| 61-144 | 244 | 63.5 |
| **Sex of the child** | | |
| Male | 203 | 52.9 |
| Female | 181 | 47.1 |
| **Residence** | | |
| Rural | 195 | 50.8 |
| Urban | 189 | 49.2 |
| **Care taker age (years)** | | |
| 20-44 | 249 | 64.8 |
| 45-59 | 77 | 20.1 |
| 60+ | 58 | 15.1 |
| **Relationship-caretaker** | | |
| Mother | 232 | 60.4 |
| Father | 32 | 8.3 |
| Grand mother | 105 | 27.3 |
| Other | 15 | 3.9 |
| **Occupation caretaker** | | |
| Employed | 91 | 23.7 |
| Business | 28 | 7.3 |
| Peasant | 150 | 39.1 |
| Un employment | 115 | 29.9 |
| **Education Caretaker** | | |
| None | 195 | 50.8 |
| Primary | 80 | 20.8 |
| Secondary/Tertiary | 109 | 28.4 |
| **Family monthly income** | | |
| <200,000 | 198 | 51.6 |
| 200,000-500,000 | 151 | 39.3 |
| >500,000 | 35 | 9.1 |
| **Siblings** | | |
| ≤2 | 214 | 55.7 |
| 3-4 | 128 | 33.3 |
| >4 | 42 | 10.9 |
| **House hold size** | | |
| ≤5 | 208 | 54.2 |
| >5 | 176 | 45.8 |
| **Birth order of participants** | | |
| 1st born | 139 | 36.2 |
| 2nd and greater | 245 | 63.8 |

**Table 2. Clinical and laboratory characteristics of study participants.**

| Characteristic | Frequency | Percentage |
|---|---|---|
| **Viral load** | | |
| Un detectable | 142 | 37.0 |
| Detected | 242 | 63.0 |
| **CD4** | | |
| <200 | 27 | 7.0 |
| 200-500 | 133 | 34.6 |
| >500 | 224 | 58.3 |
| **HIV Stage** | | |
| 1 | 299 | 77.9 |
| 2 | 65 | 16.9 |
| ≥3 | 20 | 5.2 |
| **Duration since HIV diagnosis** | | |
| ≤12 months | 120 | 31.3 |
| >12 months | 264 | 68.8 |
| **Number of admissions in preceding 6 months** | | |
| 0 | 204 | 53.1 |
| 1 | 70 | 18.2 |
| 2 | 73 | 19.0 |
| 3 | 37 | 9.6 |
| **History of Fever** | | |
| No | 353 | 91.9 |
| Yes | 31 | 8.1 |
| **ART regimen** | | |
| ABC/3TC/DTG | 327 | 85.2 |
| TDF/3TC/DTG | 49 | 12.8 |
| AZT/3TC/DTG | 8 | 2.1 |
| **Opportunistic infection** | | |
| No | 346 | 90.1 |
| Yes | 38 | 9.9 |
| **History of blood Transfusion Transfusion** | | |
| No | 364 | 94.8 |
| Yes | 20 | 5.2 |
| **History of Supplements use** | | |
| No | 361 | 94.0 |
| Yes | 23 | 6.0 |
| **History of Malaria** | | |
| Negative | 353 | 91.9 |
| Positive | 31 | 8.1 |
| **WBC** | | |
| Normal | 306 | 79.7 |
| Low | 34 | 8.9 |
| High | 44 | 11.5 |
| **RBC** | | |
| Normal | 359 | 93.5 |
| Low | 19 | 4.9 |
| High | 6 | 1.6 |

*(Continued)*

Table 2. (Continued)

| Characteristic | Frequency | Percentage |
|---|---|---|
| **Platelet count** | | |
| Normal | 325 | 84.6 |
| Low | 31 | 8.1 |
| High | 28 | 7.3 |
| **Breastfeeding duration** | | |
| <6 months | 112 | 29.2 |
| 6-12 | 236 | 61.5 |
| >12 | 36 | 9.4 |
| **Delayed introduction of complementary feeds** | | |
| No | 321 | 83.6 |
| Yes | 63 | 16.4 |
| **Exclusive breastfeeding** | | |
| No | 237 | 61.7 |
| Yes | 147 | 38.3 |
| **Wasting** | | |
| Normal | 345 | 89.8 |
| Moderate | 32 | 8.3 |
| Severe | 7 | 1.8 |
| **Stunting** | | |
| Normal | 258 | 67.2 |
| Moderate | 106 | 27.6 |
| Severe | 20 | 5.2 |
| **Underweight** | | |
| Normal | 317 | 82.6 |
| Moderate | 55 | 14.3 |
| Severe | 12 | 3.1 |
| **WaSt** | | |
| Normal | 354 | 92.2 |
| Moderate | 22 | 5.7 |
| Severe | 8 | 2.1 |

*CD4 = Cluster of differentiation; HIV = Human immune deficiency virus; ART = Anti-retroviral treatment.*

*\*WBC = white blood cell; RBC = Red blood Cells; WaSt: concurrent wasting and stunting.*

=2.014, 95% CI =(0.862–4.705) P value = **0.106**, duration of breastfeeding<6 months **(**cOR =5.211, 95% CI =(2.179–12.461) P value =**<0.001** 6–12 **(**cOR =5.577, 95% CI =(2.436–12.768) P value =**<0.001**, wastingModerate **(**cOR =1.267, 95% CI =(0.703–2.105) P value = **0.045**Severe **(**cOR =1.014, 95% CI =(0.405–2.254) P value = **0.058**, stuntingModerate **(**cOR =1.274, 95% CI = (0.806–2.015) P value = **0.300** Severe **(**cOR =6.522, 95% CI = (2.179–12.259) P value = **0.007** and WaSt**Moderate(**cOR =3.719, 95% CI = (2.358–13.164) P value = **0.005** Severe **(**cOR =1.906, 95% CI = (0.719–8.507) P value = **0.098** (Table 5).

In the multivariable analysis, lower education level of the care taker (primary) (aOR=3.204, CI = 1.596–6.829, P<0.001); having more than 4 siblings (aOR=3.428, CI = 1.376–5.212, P<0.001); having HIV stage two (aOR=2.059, CI = 1.081–3.923, P = 0.028) or three (aOR=3.093, CI = 1.303–4.946, P = 0.042); any history of admission in the preceding 6 months (aOR=1.081, CI = 1.002–3.432, P<0.001) for one admission, (aOR=1.577, CI = 1.158–4.005, P<0.001) for two admissions and (aOR=4.065, CI = 2.734–7.314, P<0.001) for three admissions; in addition to being severely wasted (aOR=2.996,

**Table 3. Prevalence of anemia among HIV positive children attended ART clinic at KRRH.**

| Characteristic | Frequency | Percentage | 95% CI |
|---|---|---|---|
| **Anemia(N = 384)** | | | |
| Yes | 220 | 57.3 | 52.3-62.2 |
| No | 164 | 42.7 | 37.8-47.7 |
| **Anemia severity N = 220** | | | |
| Mild | 47 | 21.4 | 16.4-26.4 |
| Moderate | 107 | 48.6 | 42.3-55.5 |
| Severe | 66 | 30.0 | 23.6-35.9 |
| **Anemia type (MCV) N = 220** | | | |
| Normocytic | 20 | 9.1 | 5.5-12.7 |
| Microcytic | 181 | 82.3 | 77.3-86.8 |
| Macrocytic | 19 | 8.6 | 5.5-12.7 |
| **Anemia type (MCH) N = 220** | | | |
| Normochromic | 24 | 10.9 | 6.8-15.0 |
| Hypochromic | 196 | 89.1 | 85.0-93.2 |

*MCV = mean corpuscular volume or brother, MCH = mean corpuscular hemoglobin.*

CI = 1.262–9.781, P = 0.022), severely stunted (aOR=3.847, CI = 2.063–7.190, P = 0.009), or having concurrent stunting and wasting (aOR=3.003, CI = 1.972–12.138, P = 0.001) were independently associated with anemia among children with HIV (Table 6).

## Discussion

### Prevalence of anemia among HIV positive children at KRRH

Regarding, the prevalence, 57.3% of the participants had anemia, which was predominantly moderate according to severity with Microcytosis and hypochromia. This prevalence was high. This high prevalence can be explained by the low social economic status of the participants as shown by the family monthly income, which was less than 200,000 in over 50% of the participants (Table 1). The low income might be affecting the feeding practices, resulting in nutritional deficiencies [12]. These nutritional deficiencies can often contribute to the anemia [13], as seen in this study where stunting and wasting all had a significant association with anemia.

It is also important to note that, undernutrition is often associated with micronutrient deficiencies [14]. This can be affirmed by the fact that majority of the participates in the current study had microcytic anemia (Table 3), whose main etiology is Iron deficiency. Another indicator of low social economic status is the education level, where over 70% of the caretakers either never had a formal education or only had a primary education (Table 1). The low education level generally affects the financial status as well as the understandingin regard to the care and nutrition of children. The poor social economic status can also result in comorbidities like intestinal warms which could also results in recurrent blood loss and anemia as reported by [15] in Ethiopia.

Another possible contributor to the high prevalence could have been inadequate breastfeeding. The reduced duration of breastfeeding has been shown to increase risk of childhood infections, some of which increase the risk of anemia [16]. Also inadequate breastfeeding increases the risk of malnutrition [12], which in turn increases risk of anemia as explained in the previous paragraphs.

The prevalence reported was comparable to that reported among the 164 children in Lagos, Nigeria, who were between the ages of 5 and 12 years 54.2% HIV-positive children were anemic [17], a study done in Mulago by Munyagwa et al which found that 50.7% of HIV positive children were anemic [6] as well as a study done in western Uganda at

**Table 4. Bivariable analysis of sociodemographic factors associated with anemia among HIV positive children attended antiretroviral treatment clinic at KRRH. more likely to have blood culture positive sepsiss (aOR=red patient who had a d the symptoms was 13.**

| Characteristic | No Anemia, N=164 | Anemia, N=220 | cOR | 95% CI | P value |
|---|---|---|---|---|---|
| **Age (months)** | | | | | |
| 6-24 | 19(11.6) | 26(11.8) | Ref | | |
| 25-36 | 15(9.1) | 31(14.1) | 1.510 | 0.643-3.549 | 0.344 |
| 37-60 | 37(22.6) | 12(5.5) | 0.237 | 0.098-0.571 | **0.001** |
| 61-144 | 93(56.7) | 151(68.6) | 1.187 | 0.622-2.263 | 0.604 |
| **Sex of the child** | | | | | |
| Male | 85(51.8) | 118(53.6) | 1.075 | 0.717-1.612 | 0.726 |
| Female | 79(48.2) | 102(46.4) | Ref | | |
| **Residence** | | | | | |
| Rural | 69(42.1) | 126(57.3) | 1.846 | 1.226-2.779 | **0.003** |
| Urban | 95(57.9) | 94(42.7) | Ref | | |
| **Caretaker's age (years)** | | | | | |
| 20-44 | 98(59.8) | 151(68.6) | 1.438 | 0.810-2.554 | 0.215 |
| 45-59 | 38(23.2) | 39(17.7) | 0.958 | 0.484-1.894 | 0.902 |
| 60+ | 28(17.1) | 30(13.6) | Ref | | |
| **Relationship-caretaker** | | | | | |
| Mother | 89(54.3) | 143(65) | Ref | | |
| Father | 24(14.6) | 8(3.6) | 0.207 | 0.089-0.482 | **<0.001** |
| Grand mother | 40(24.4) | 65(29.5) | 1.011 | 0.629-1.625 | 0.963 |
| Other** | 11(6.7) | 4(1.8) | 0.226 | 0.070-0.733 | **0.013** |
| **Occupation caretaker** | | | | | |
| Employed | 37(22.6) | 54(24.5) | Ref | | |
| Business | 12(7.3) | 16(7.3) | 0.914 | 0.388-2.153 | 0.836 |
| Peasant | 64(39.0) | 86(39.1) | 0.921 | 0.543-1.562 | 0.759 |
| Un employment | 51(31.1) | 64(29.1) | 0.860 | 0.493-1.501 | 0.595 |
| **Education Caretaker** | | | | | |
| None | 93(56.7) | 102(46.4) | 0.863 | 0.539-1.382 | 0.540 |
| Primary | 23(14.0) | 57(25.9) | 1.950 | 1.055-3.605 | **0.033** |
| Secondary/Tertiary | 48(29.3) | 61(27.7) | Ref | | |
| **Family monthly income** | | | | | |
| <200,000 | 83(50.6) | 115(52.3) | 2.656 | 1.251-5.638 | **0.011** |
| 200,000-500,000 | 58(35.4) | 93(42.3) | 3.073 | 1.421-6.645 | **0.004** |
| >500,000 | 23(14.0) | 12(5.5) | Ref | | |
| **Number of siblings** | | | | | |
| ≤2 | 97(59.1) | 117(53.2) | Ref | | |
| 3-4 | 63(38.4) | 65(29.5) | 0.855 | 0.552-1.327 | 0.485 |
| >4 | 4(2.4) | 38(17.3) | 7.876 | 2.716-22.844 | **<0.001** |
| **House hold size** | | | | | |
| ≤5 | 80(48.8) | 128(58.2) | Ref | | |
| >5 | 84(51.2) | 92(41.8) | 0.685 | 0.456-1.028 | 0.216 |
| **Birth order** | | | | | |
| 1st born | 65(39.6) | 74(33.6) | 0.772 | 0.507-1.175 | 0.227 |
| 2nd and greater | 99(60.4) | 146(66.4) | Ref | | |

*=aunt, grandfather or brother, cOR= Crude odds ratio, CI=Confidence interval, Ref=reference category.

**Table 5.** Bivariable analysis of clinical and laboratory factors associated with anemia among HIV positive children attended antiretroviral treatment clinic at KRRH. more likely to have blood culture positive sepsiss (aOR=red patient who had a d the symptoms was 15.

| Characteristic | No Anemia, N=164 | Anemia, N=220 | cOR | 95% CI | P value |
|---|---|---|---|---|---|
| **Viral load** | | | | | |
| Un detectable | 54(32.9) | 88(40.0) | Ref | | |
| Detected | 110(67.1) | 132(60.0) | 0.736 | 0.482-1.124 | 0.256 |
| **CD4** | | | | | |
| <200 | 19(11.6) | 8(3.6) | 0.147 | 0.061-1.354 | 0.214 |
| 200-500 | 87(53.0) | 46(20.9) | 0.185 | 0.116-1.294 | 0.234 |
| >500 | 58(35.4) | 166(75.5) | Ref | | |
| **HIV Stage** | | | | | |
| 1 | 141(86.0) | 158(71.8) | Ref | | |
| 2 | 19(11.6) | 46(20.9) | 2.161 | 1.209-3.862 | **0.009** |
| ≥3 | 4(2.4) | 16(7.3) | 3.570 | 1.166-10.929 | **0.026** |
| **Duration since HIV diagnosis** | | | | | |
| ≤12 months | 51(31.1) | 69(31.4) | Ref | | |
| >12 months | 113(68.9) | 151(68.6) | 0.988 | 0.638-1.528 | 0.956 |
| **Number of admissions in preceding 6 months** | | | | | |
| 0 | 114(69.5) | 90(40.9) | Ref | | |
| 1 | 31(18.9) | 39(17.7) | 1.594 | 0.923-2.753 | **0.095** |
| 2 | 12(7.3) | 61(27.7) | 6.439 | 3.269-12.682 | **<0.001** |
| 3 | 7(4.3) | 30(13.6) | 5.429 | 2.279-12.930 | **<0.001** |
| **History of Fever** | | | | | |
| No | 149(90.9) | 204(92.7) | Ref | | |
| Yes | 15(9.1) | 16(7.3) | 0.779 | 0.373-1.625 | 0.506 |
| **ART regimen** | | | | | |
| ABC/3TC/DTG | 145(88.4) | 182(82.7) | Ref | | |
| TDF/3TC/DTG | 18(11.0) | 31(14.1) | 1.372 | 0.738-2.552 | 0.318 |
| AZT/3TC/DTG | 1(0.6) | 7(3.2) | 5.577 | 0.678-45.844 | **0.110** |
| **Opportunistic infection** | | | | | |
| No | 149(90.9) | 197(89.5) | Ref | | |
| Yes | 15(9.1) | 23(10.5) | 1.160 | 0.585-2.299 | 0.671 |
| **History of Malaria** | | | | | |
| Negative | 149(90.9) | 204(92.7) | Ref | | |
| Positive | 15(9.1) | 16(7.3) | 0.779 | 0.373-1.625 | 0.506 |
| **WBC** | | | | | |
| Normal | 145(88.4) | 161(73.2) | Ref | | |
| Low | 11(6.7) | 23(10.5) | 1.883 | 0.887-3.997 | **0.099** |
| High | 8(4.9) | 36(16.4) | 4.053 | 1.824-9.004 | **0.001** |
| **Platelet count** | | | | | |
| Normal | 145(88.4) | 180(81.8) | Ref | | |
| Low | 11(6.7) | 20(9.1) | 1.465 | 0.680-3.156 | 0.330 |
| High | 8(4.9) | 20(9.1) | 2.014 | 0.862-4.705 | **0.106** |
| **Breastfeeding duration** | | | | | |
| <6 months | 45(27.4) | 67(30.5) | 5.211 | 2.179-12.461 | **<0.001** |
| 6-12 | 91(55.5) | 145(65.9) | 5.577 | 2.436-12.768 | **<0.001** |
| >12 | 28(17.1) | 8(3.6) | Ref | | |

*(Continued)*

**Table 5.** (Continued)

| Characteristic | No Anemia, N=164 | Anemia, N=220 | cOR | 95% CI | P value |
|---|---|---|---|---|---|
| **Delayed introduction of complementary feeds** | | | | | |
| No | 136(82.9) | 185(84.1) | Ref | | |
| Yes | 28(17.1) | 35(15.9) | 0.919 | 0.533-1.583 | 0.761 |
| **Exclusive breastfeeding** | | | | | |
| No | 91(55.5) | 146(66.4) | Ref | | |
| Yes | 73(44.5) | 74(33.6) | 0.632 | 0.417-1.958 | 0.216 |
| **Wasting** | | | | | |
| Normal | 149(90.8) | 196(89.1) | Ref | | |
| Moderate | 12(7.3) | 20(9.1) | 1.267 | 0.703-2.105 | **0.045** |
| Severe | 3(1.8) | 4(1.8) | 1.014 | 0.405-2.254 | **0.058** |
| **Stunting** | | | | | |
| Normal | 120(73.2) | 138(62.7) | Ref | | |
| Moderate | 43(26.2) | 63(28.6) | 1.274 | 0.806-2.015 | **0.300** |
| Severe | 1(0.6) | 19(8.6) | 6.522 | 2.179-12.259 | **0.007** |
| **Underweight** | | | | | |
| Normal | 136(82.9) | 181(82.3) | Ref | | |
| Moderate | 4(14.6) | 31(14.1) | 0.971 | 0.545-1.729 | 0.919 |
| Severe | 4(2.4) | 8(3.6) | 1.503 | 0.443-5.094 | 0.513 |
| **WaSt** | | | | | |
| Normal | 162(98.8) | 192(87.3) | Ref | | |
| Moderate | 1(0.6) | 21(9.5) | 3.719 | 2.358-13.164 | **0.005** |
| Severe | 1(0.6) | 7(3.2) | 1.906 | 0.719-8.507 | **0.098** |

*=aunt, grandfather or brother, cOR= Crude odds ratio, CI=Confidence interval, CD4=Cluster of differentiation; HIV=Human immune deficiency virus; ART=Anti-retroviral treatment.

*=aunt, grandfather or brother, cOR= Crude odds ratio, CI=Confidence interval; WBC=White blood count; RBC=Red blood count; WaSt=concurrent wasting and stunting, Ref=reference category.

Mbarara Regional Referral Hospital where anemia was present in 57.6% of the HIV positive children [7]. The similarities can be explained by the comparability of the study populations (same sex, same age group) and study design (cross-sectional). However, unlike those studies, the current research in Kayunga provides additional insight into the burden of iron deficiency anemia; an aspect previously underexplored in Ugandan pediatric HIV populations. This is particularly relevant given Uganda's National Anaemia Policy (2002), which recommends routine screening and preventive treatment for anemia in HIV-positive children. Despite these guidelines, implementation remains inconsistent across regional hospitals, especially in semi-rural districts like Kayunga. The high prevalence of anemia (57.3%) and iron deficiency anemia (54.1% among anemic children) observed suggests gaps in operationalizing these national recommendations.

The prevalence reported was higher than that reported by a meta-analysis of 63 observational studies encompassing 110,113 HIV positive children where anemia was seen in 39.7% [18]. The difference was possibly because of the geographical area, lower sample size, participants age group and study design [19], this could explain the higher prevalence in our study. The same could explain the reason for the difference in prevalence reported in Walaito, Ethiopia, among 256 children from 6 months to 14 years of age with a prevalence of 38.8% [20].

The prevalence reported was lower than that reported in Mozambique, where 88% of the HIV positive children were anemic [21]. The explanation for the difference is because the Mozambique study only enrolled hospitalized children who had a documented fever or history of fever within the 24 hours before hospital admission. As seen in our study, history of

**Table 6. Multivariable analysis of factors associated with anemia among HIV positive children attended ART clinic at KRRH.**

| Characteristic | cOR | 95% CI | P value | aOR | 95% CI | P value |
|---|---|---|---|---|---|---|
| **Age of the child** | | | | | | |
| 6-24 | Ref | | | | | |
| 25-36 | 1.510 | 0.643-3.549 | 0.344 | 0.991 | 0.244-4.031 | 0.990 |
| 37-60 | 0.237 | 0.098-0.571 | 0.001 | 0.553 | 0.155-1.976 | 0.362 |
| 61-144 | 1.187 | 0.622-2.263 | 0.604 | 0.472 | 0.162-1.371 | 0.168 |
| **Residence** | | | | | | |
| Rural | 1.846 | 1.226-2.779 | 0.003 | 1.407 | 0.538-3.682 | 0.486 |
| Urban | Ref | | | | | |
| **Relationship-caretaker** | | | | | | |
| Mother | Ref | | | | | |
| Father | 0.207 | 0.089-0.482 | <0.001 | 0.086 | 0.027-1.270 | 0.051 |
| Grand mother | 1.011 | 0.629-1.625 | 0.963 | 1.154 | 0.632-2.108 | 0.366 |
| Other** | 0.226 | 0.070-0.733 | 0.013 | 0.195 | 0.053-1.717 | 0.155 |
| **Education of Caretaker** | | | | | | |
| None | 0.863 | 0.539-1.382 | 0.540 | 1.234 | 0.679-2.240 | 0.490 |
| **Primary** | **1.950** | **1.055-3.605** | **0.033** | **3.204** | **1.596-6.829** | **<0.001** |
| Secondary/Tertiary | Ref | | | | | |
| **Family monthly income** | | | | | | |
| <200,000 | 2.656 | 1.251-5.638 | 0.011 | 0.452 | 0.099-2.062 | 0.305 |
| 200,000-500,000 | 3.073 | 1.421-6.645 | 0.004 | 2.541 | 0.496-13.000 | 0.263 |
| >500,000 | Ref | | | | | |
| **Number of siblings** | | | | | | |
| ≤2 | Ref | | | | | |
| 3-4 | 0.855 | 0.552-1.327 | 0.485 | 0.571 | 0.324-1.007 | 0.053 |
| >4 | **7.876** | **2.716-22.844** | **<0.001** | **3.428** | **1.376-5.212** | **<0.001** |
| **HIV Stage** | | | | | | |
| 1 | Ref | | | | | |
| 2 | **2.161** | **1.209-3.862** | **0.009** | **2.059** | **1.081-3.923** | **0.028** |
| ≥3 | **3.570** | **1.166-10.929** | **0.026** | **3.093** | **1.303-4.946** | **0.042** |
| **Number of admissions in preceding 6 months** | | | | | | |
| 0 | Ref | | | | | |
| 1 | **1.594** | **0.923-2.753** | **0.095** | **1.081** | **1.002-3.432** | **<0.001** |
| 2 | **6.439** | **3.269-12.682** | **<0.001** | **1.577** | **1.158-4.005** | **<0.001** |
| 3 | **5.429** | **2.279-12.930** | **<0.001** | **4.065** | **2.734-7.314** | **<0.001** |
| **ART regimen** | | | | | | |
| ABC/3TC/DTG | Ref | | | | | |
| TDF/3TC/DTG | 1.372 | 0.738-2.552 | 0.318 | 1.533 | 0.443-5.312 | 0.500 |
| AZT/3TC/DTG | 5.577 | 0.678-45.844 | 0.110 | 0.562 | 0.042-7.614 | 0.665 |
| **WBC** | | | | | | |
| Normal | Ref | | | | | |
| Low | 1.883 | 0.887-3.997 | 0.099 | 0.142 | 0.014-1.400 | 0.095 |
| High | 4.053 | 1.824-9.004 | 0.001 | 0.963 | 0.248-3.732 | 0.956 |
| **Platelet count** | | | | | | |
| Normal | Ref | | | | | |
| Low | 1.465 | 0.680-3.156 | 0.330 | 7.285 | 0.856-62.025 | 0.069 |
| High | 2.014 | 0.862-4.705 | 0.106 | 1.637 | 0.441-6.075 | 0.461 |

*(Continued)*

**Table 6.** (Continued)

| Characteristic | cOR | 95% CI | P value | aOR | 95% CI | P value |
|---|---|---|---|---|---|---|
| **Breastfeeding duration** | | | | | | |
| <6 months | 5.211 | 2.179-12.461 | <0.001 | 1.291 | 0.326-5.115 | 0.717 |
| 6-12 | 5.577 | 2.436-12.768 | <0.001 | 2.592 | 0.679-9.895 | 0.163 |
| >12 | Ref | | | | | |
| **Wasting** | | | | | | |
| Normal | Ref | | | | | |
| Moderate | 1.267 | 0.703-2.105 | 0.025 | 1.378 | 0.701-2.413 | 0.065 |
| Severe | **1.014** | **0.405-2.254** | **0.037** | **1.224** | **0.262-2.542** | **0.023** |
| **Stunting** | | | | | | |
| Normal | Ref | | | | | |
| Moderate | 1.274 | 0.806-2.015 | 0.300 | 0.408 | 0.211-1.790 | 0.088 |
| Severe | **6.522** | **2.179-12.259** | **0.007** | **3.847** | **2.063-7.190** | **0.009** |
| **WaSt** | | | | | | |
| Normal | Ref | | | | | |
| Moderate | **3.719** | **2.358-13.164** | **0.005** | **3.003** | **1.972-12.138** | **0.001** |
| Severe | 1.906 | 0.719-8.507 | 0.098 | 1.093 | 0.029-14.535 | 0.962 |

*=aunt, grandfather or brother, aOR= adjusted odds ratio, Ref=reference category.

hospitalization increases odds of anemia (Table 5). Also, history of fever is suggestive of infections such as malaria that increase risk of anemia [22]. The fact that our study enrolled children who were stable coming to the hospital for routine care as opposed to the Mozambique study which enrolled admitted children with fever could explain the difference.

The prevalence in our study was also lower than the one done in the pediatric department of Mulago Hospital, which comprised 39 uninfected and 165 HIV-positive 9-month-old infants where the prevalence was 76.9% in the uninfected group and 90.9 in the HIV-positive group [23]. The possible explanations for the differences include the fact that the Mulago study was limited to children aged 9 months who curry a higher risk for anemia [24] as well as the fact that some of the HIV positive infants were not on ART, yet ART has been associated with a lower risk of anemia [5].

### Factors associated with anemia among HIV positive children at KRRH

Regarding factors, children whose caretakers had a lower education level, were 3.204 times more likely to have anemia. Children who had more than four siblings were 3.428 times more likely to have anemia. The education level and number of siblings all are indicators of social economic status. In most of the setting in Uganda, a lower education is associated with a lower income, while a high number of children (siblings) also is often associated with poor social economic status, since the limited resources available, are to be divided among the many children resulting in subpar feeding and care for each of the children; which results in undernutrition and anemia [25,26].

Having HIV stage two increased odds of anemia by 2.059 times, while the odds were increased by 3.093 for those with stage three. This was in agreement with findings by [10] who reported that the risk of anemia increased as HIV-clinical stage increased [10]. A higher HIV stage increases risk of opportunistic infections as well as the possibility of bone marrow suppression which in turn increase the risk of anemia [5]. The same can explain why a history of admission in the preceding 6 months was associated with higher odds for having anemia. The odds were increased by 1.081 times for one admission, by 1.577 times for two admissions and by 4.065 times for three admissions in agreement with finding by [21] in Mozambique.

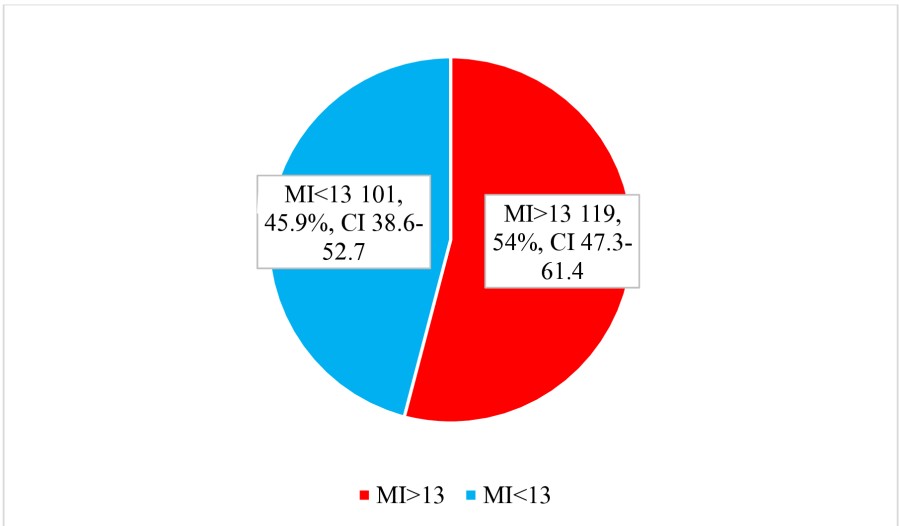

**Fig 1. Proportion of IDA diagnosed by Mentzer Index >13 among anemic children.** Iron deficiency anemia accounted for 54.1% of all children with anemia. In consideration of the total number of enrolled children (384), the overall prevalence of Iron deficiency anemia was 31.0%. Details are shown Fig 1.

Children who were severely wasted were 1.224times more likely to have anemia compared to those who had a normal nutrition status, while the odds of anemia were increased by 3.847 times among those who were severely stunted and by 3.003 times among those who had concurrent stunting and wasting. Stunting is an indicator of chronic malnutrition while wasting indicates acute malnutrition. With malnutrition, comes a micronutrient deficiency, resulting in an increased risk of anemia. This was in agreement with findings in India, where malnutrition was found to be an important risk factor of anemia [24], in Burkina Faso where results showed a significant correlation between BMI and anemia [19], in Mozambique, where anemic children had a higher risk of undernutrition [22], in Ethiopia, where having undernutrition was an independent predictor of anemia in children undergoing ART [9] and at Mulago Hospital, where a substantial correlation was found between moderate anemia and height for age Z-score <2 [1].

The study's identification of caregiver education level, advanced HIV clinical stage, number of siblings, admission history, stunting and wasting as key predictors of anemia among HIV-positive children aligns with Uganda's broader health and nutrition policy frameworks, yet also reveals critical implementation gaps. The Maternal, Infant, Young Child and Adolescent Nutrition (MIYCAN) Guidelines (2021) emphasize the importance of caregiver education and community-based nutrition interventions to combat undernutrition and micronutrient deficiencies in vulnerable populations. Similarly, Uganda's National Anaemia Policy and HIV/AIDS Strategic Plan advocate for integrated screening and management of anemia within HIV care, particularly for pediatric populations. However, the high prevalence of anemia and iron deficiency anemia observed in this semi-rural setting suggests that these policies are not being fully operationalized at the district level. The association with advanced HIV staging further reveals the need for strengthened adherence support, early ART initiation, and routine hematologic monitoringcomponents emphasized in national HIV guidelines but inconsistently implemented across decentralized health facilities. The link between wasting and anemia reinforces the urgency of scaling up nutrition-sensitive HIV programming, as outlined in Uganda's Nutrition Action Plan, which calls for multisectoral coordination to address both acute and chronic malnutrition.

## Study limitations

While the study found that 54.1% of anemic children had iron deficiency anemia based on the MI > 13, this estimate may be inflated due to the inherent limitations of relying solely on this hematological ratio. The MI, calculated as MCV

divided by RBC count, is a useful screening tool in resource-limited settings but lacks specificity and sensitivity compared to biochemical markers. It cannot distinguish between iron deficiency and other causes of microcytic anemia such as anemia of chronic disease or early thalassemia trait. Secondly, this study employed a facility-based sampling approach, enrolling HIV-positive children attending the ART clinic at Kayunga Regional Referral Hospital. While this method ensured access to clinical records and standardized laboratory procedures, it may limit the generalizability of findings to the broader pediatric HIV population in Uganda. Children who regularly attend ART clinics may differ systematically from those in the community who are lost to follow-up, not yet enrolled in care, or receiving services from lower-level health facilities. These groups may have differing nutritional profiles, socioeconomic conditions, or disease severity, potentially introducing selection bias. As a result, the prevalence and associated factors of anemia reported here may not fully reflect the burden among HIV-positive children in remote or underserved settings. Due to budget constraints, diagnostic studies for iron deficiency, such as serum iron, ferritin, and total iron-binding capacity (TIBC), were not conducted. Additionally, other potential contributors to anemia in HIV, including vitamin B12 and folic acid deficiencies, were not investigated. This may have limited the study's ability to fully determine the underlying causes of anemia in the study population.

## Conclusion

The prevalence of anemia was high and this was determined by Lower education level of the caretaker, having more than four siblings, having HIV stages 2–3, any history of admission in the preceding 6 months and being severely stunted, wasted or concurrent stunting and wasting.

Routine screening for anemia should be integrated into existing ART clinic workflows, using point-of-care hemoglobin testing to enable early detection and management. In addition, leveraging community health workers (CHWs) offers a feasible and scalable strategy for caregiver education and follow-up. CHWs, already embedded within Uganda's decentralized health system, can be trained to deliver targeted messages on nutrition, iron-rich diets, and adherence to ART and supplementation regimens. They can also facilitate referrals for children with suspected anemia and monitor treatment compliance during home visits. School-based health programs and local radio campaigns may further reinforce awareness, especially in communities with low literacy levels. Strengthening supply chains for iron supplements and multivitamins, alongside capacity-building for frontline health workers, would enhance the sustainability of these interventions. By aligning with Uganda's Health Sector Development Plan (HSDP) and Community Health Extension Worker Strategy, these approaches offer practical pathways to reduce anemia burden and improve child health outcomes in underserved regions. Future research should prioritize longitudinal cohort studies to explore the causal relationships between HIV disease progression and anemia in pediatric populations.

## Author contributions

**Conceptualization:** Abdirashid Dahir Herow, Ndeezi Grace, Martin Nduwimana.

**Data curation:** Abdirashid Dahir Herow, Ndeezi Grace, Joel Kimera, Martin Nduwimana.

**Formal analysis:** Abdirashid Dahir Herow.

**Investigation:** Abdirashid Dahir Herow.

**Methodology:** Abdirashid Dahir Herow, Ndeezi Grace, Martin Nduwimana.

**Project administration:** Abdirashid Dahir Herow.

**Supervision:** Ndeezi Grace, Joel Kimera, Martin Nduwimana.

**Validation:** Abdirashid Dahir Herow, Ndeezi Grace, Joel Kimera, Martin Nduwimana.

**Visualization:** Abdirashid Dahir Herow, Ndeezi Grace, Joel Kimera, Martin Nduwimana.

**Writing – original draft:** Abdirashid Dahir Herow.

**Writing – review & editing:** Abdirashid Dahir Herow.

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
