## [Decision Letter · Decision Letter 0]

23 Jun 2025

PONE-D-25-17782Prevalence and Factors Associated With Anemia among HIV-Positive Children in Central Uganda: A Cross-sectional studyPLOS ONE

Dear Dr. Herow,

Thank you for submitting your manuscript to PLOS ONE. After careful consideration, we feel that it has merit but does not fully meet PLOS ONE’s publication criteria as it currently stands. Therefore, we invite you to submit a revised version of the manuscript that addresses the points raised during the review process. Please submit your revised manuscript by Aug 07 2025 11:59PM. If you will need more time than this to complete your revisions, please reply to this message or contact the journal office at plosone@plos.org . Please include the following items when submitting your revised manuscript:

We look forward to receiving your revised manuscript.

Kind regards,

Alqeer Aliyo Ali, MSc

Academic Editor

PLOS ONE

Journal Requirements:

2. We note that your Data Availability Statement is currently as follows: All relevant data are within the manuscript and in Supporting Information files.

5. Please upload a copy of Figure 3, to which you refer in your text on page 18. If the figure is no longer to be included as part of the submission please remove all reference to it within the text.

Additional Editor Comments:

** ** The reviewers completed their review and attached it to this letter. The author should revise the title, abstract, introduction, methodology, results, and discussion as per the reviewers, comments. Also, the author(s) should make sure that the revised manuscript format aligns with the journal guideline. 

Reviewers' comments:

Reviewer's Responses to Questions

**Comments to the Author**

1. Is the manuscript technically sound, and do the data support the conclusions?

Reviewer #1: Partly

Reviewer #2: Yes

2. Has the statistical analysis been performed appropriately and rigorously? 

Reviewer #1: No

Reviewer #2: Yes

3. Have the authors made all data underlying the findings in their manuscript fully available?

Reviewer #1: Yes

Reviewer #2: Yes

4. Is the manuscript presented in an intelligible fashion and written in standard English?

Reviewer #1: Yes

Reviewer #2: Yes

5. Review Comments to the Author

Reviewer #1: I have reviewed the manuscript and it looks interesting. However, I found that the abstract, introduction, methodology, results, and discussions needs revisions before consideration for next steps. Therefore, I recommend the author (s) should revise the manuscript as per attached comments.

Reviewer #2: I have reviewed the manuscript and I need these revisions will enhance clarity, accuracy, and adherence to journal guidelines.

The title in the manuscript draft ("Prevalence and Factors Associated With Anemia among HIV-Positive Children in Central Uganda: A Cross-sectional study") differs slightly from the title in the cover letter ("Prevalence and Factors Associated with Anemia among HIV-Positive Children at Kayunga Regional Referral Hospital, Uganda"). Ensure consistency throughout the document.

The abstract mentions "Mentzer index >13" for diagnosing iron deficiency anemia (IDA), but the Results section states "MI equal to or more than 13." Clarify the exact cutoff used (e.g., ">13" or "≥13") for consistency.

The sample size calculation cites a prevalence of 50.7% from a study by Munyagwa et al. (Ref 6), but the reference list does not include this study. Provide the full citation or clarify the source.

The manuscript states data was collected between November 2024 and January 2025, which appears to be a future date. Verify and correct the timeline (e.g., 2023–2024).

The Methods section mentions using SPSS for logistic regression, but the Results section states STATA was used. Clarify which software was employed and ensure consistency.

Tables 4–6 reference "cOR" (crude odds ratio) and "aOR" (adjusted odds ratio), but these abbreviations are not defined in the text or list of abbreviations. Add definitions for clarity.

Figure 1 (Iron deficiency anemia) is referenced in the text but lacks a descriptive caption in the manuscript. Include a clear caption explaining the pie chart (e.g., "Proportion of IDA diagnosed by Mentzer Index >13 among anemic children").

The ethics approval number ("KIU-2024-517") suggests approval in 2024, but the data collection period is also listed as 2024–2025. Confirm the approval date aligns with the study timeline.

References 5 and 26 are incomplete or unclear. Reference 5 cites "NEMA (2008)" without a title, and Reference 26 lacks authors. Provide full details for all references.

The Limitations section notes that serum iron, ferritin, and TIBC were not measured due to budget constraints. Consider acknowledging how this might affect the Mentzer index's reliability as a surrogate for IDA diagnosis.

Additional Notes

Ensure all abbreviations (e.g., MI, IDA) are defined at first use.

Verify numerical consistency (e.g., "54.1% of children with anemia" vs. "31.0% overall prevalence of IDA").

Check for typographical errors (e.g., "care taker" vs. "caretaker").

6. PLOS authors have the option to publish the peer review history of their article (what does this mean? ). If published, this will include your full peer review and any attached files.

**Do you want your identity to be public for this peer review?** For information about this choice, including consent withdrawal, please see our Privacy Policy .

Reviewer #1: No

Reviewer #2: No

---

## [Author Response · Author response to Decision Letter 1]

30 Jul 2025

I am grateful for the opportunity to revise and enhance the quality of my manuscript in response to the feedback provided.

Comment by reviewer: The title in the manuscript draft ("Prevalence and Factors Associated With Anemia among HIV-Positive Children in Central Uganda: A Cross-sectional study") differs slightly from the title in the cover letter ("Prevalence and Factors Associated with Anemia among HIV-Positive Children at Kayunga Regional Referral Hospital, Uganda"). Ensure consistency throughout the document.

Response by the author: The title has been amended to reflect the scope of the study more clearly. The new title is: Prevalence and Factors Associated with Anemia among HIV-Positive Children in Central Uganda: A Cross-Sectional Study

Abstract

Comment by the reviewer: In the abstract, the sentence “Anemia was diagnosed if the hemoglobin level was below 11 g/dl…” should be rephrased for clarity.

Response by author: this has been rephrased and clearly stated (The diagnosis of anaemia was made if the haemoglobin level was less than 11g/dl)

Comment by the reviewer: The phrase “lower education level of the caretaker” lacks specificity. Clarify whether this refers to primary education or no formal education.

Response by the author: this has been clarified and used primary level of education and non formal education were independently associated with anemia

Introduction

Comment by the reviewer: Global vs. Local Context: The introduction provides global and regional data well but could more explicitly state how this study fills a gap in the literature specific to Kayunga, beyond other Ugandan regions like Mbarara or Mulago.

Response by the author: This has been revised

Methodology

Comment by the reviewer: Sampling Justification: Please justify why a sample size of 384 was chosen (e.g., power calculation). Also clarify if this number was based on expected effect size or previous prevalence estimates.

Response by the author: sample size calculation was used KishLeslie formula to get sample size while using previous prevalence estimated and chosen power calculation

Comment by the reviewer: Ethical Statement: It would enhance transparency to mention that written informed consent was obtained from caregivers and assent from older children in the Methods section, not only in the ethics statement.

Response by the author: this has been mentioned in method section

Comment by the reviewer: Data Collection Tools: Were the structured questionnaires pilot-tested before use? If so, include this information to strengthen methodological rigor.

Response by the author: yes, this has been stated

Comment by the reviewer: Mentzer Index Calculation: Define the Mentzer index formula (MCV/RBC ratio) in the Methods section for readers unfamiliar with it.

Response by the author: this has been defined in the method section

Comment by the reviewer Statistical Analysis Software Mention: While SPSS was used for logistic regression; STATA was mentioned later for data analysis. Clarify which software was primarily used and for what parts.

Response by the author: this has been corrected and SPSS was used for logistic regression

Results Section Feedback

Comment by the reviewer: Severe Stunting/Wasting Definitions: Clarify how stunting and wasting were classified as "severe" were these based on WHO Z-scores (< -3 SD)? This should be clearly stated.

Response by the author: this has been clearly stated and mention in the method section that has been used WHO growth chart (WHO Z-scores (<-3SD)

Discussion and Conclusion

Comment by the reviewer: Comparison with Previous Studies: The discussion compares findings with studies in Ethiopia, Nigeria, and India, but less attention is given to local Ugandan contexts. Expand on how findings differ or align with national-level policies or programs.

Response by the author: this has been revised

Minor Editorial Suggestions

Comment by the editorial: Check all acronyms for first-time definitions (e.g., MI = Mentzer Index).

Response by the author: all acronyms has be revised and defined first time MI

---

## [Editor Report · Decision Letter 1]

3 Aug 2025

PONE-D-25-17782R1Prevalence and Factors Associated With Anemia among HIV-Positive Children in Central Uganda: A Cross-sectional studyPLOS ONE

Dear Dr. Herow,

Thank you for submitting your manuscript to PLOS ONE. After careful consideration, we feel that it has merit but does not fully meet PLOS ONE’s publication criteria as it currently stands. Therefore, we invite you to submit a revised version of the manuscript that addresses the points raised during the review process.

**ACADEMIC EDITOR: ** Although the author responded to and revised the reviewers' queries and comments, the majority of them were not corrected in the manuscript.

The reviewers' comments on the introduction were not incorporated into the revised version of the manuscript.The methodology section should be structured as follows: Study setting and period, study population, inclusion and exclusion criteria (eligibility), sample size determination, sampling techniques, data collection method, operational definitions/terms, data quality control (assurance), ethical consideration/approval (including reference number and the name of the approved organization/institution), and data analysis.Some of the reviewers' comments on the results section have not been corrected.The reviewers' comments on the discussion were not considered in the revised version of the manuscript. Therefore, I strongly recommend that the author(s) clearly revise the manuscript in accordance with the previous reviewers' comments. 

We look forward to receiving your revised manuscript.

Kind regards,

Alqeer Aliyo Ali, MSc

Academic Editor

PLOS ONE
---

## [Author Response · Author response to Decision Letter 2]

12 Sep 2025

Reviewer one comments: Responses

Thank you for providing me with the opportunity to review the manuscript titled "Prevalence and Factors Associated with Anemia among HIV-Positive Children in Central Uganda: A Cross-sectional Study." While the study is important for researchers and stakeholders, I have a concern that could help improve the manuscript:

General Comments and Suggestions

1. The title is informative but can be slightly shortened without losing meaning. Consider:

“Prevalence and Determinants of Anemia among HIV-Positive Children in Central Uganda: A Cross-Sectional Study.”

Dear reviewer, thank you so much for taking time to go through our manuscript. As advised, we have slightly changed the title of the paper to “Prevalence and Determinants of Anemia among HIV-Positive Children in Central Uganda: A Cross-Sectional Study.”

Abstract

2. In the abstract, the sentence “Anemia was diagnosed if the hemoglobin level was below 11 g/dl…” should be rephrased for clarity.

Dear Reviewer, We have improved on the sentence by making it more clear. It has been changed to “Anemia was defined based on age-specific hemoglobin thresholds: <11.0 g/dL for children aged 6–59 months, <11.5 g/dL for those aged 5–11 years, and <12.0 g/dL for children aged 12–14 years.”

3. The phrase “lower education level of the caretaker” lacks specificity. Clarify whether this refers to primary education or no formal education.

Dear reviewer, we have changed “lower education level of the caretaker to “caretakers who stopped in primary.”

4. Keywords: Consider adding “pediatrics” , “nutritional deficiency” , and “cross-sectional study” to improve discoverability.

We have added the key words pediatrics, nutritional deficiency, cross sectional study to the manuscript; just after the abstract.

Introduction

5. Global vs. Local Context: The introduction provides global and regional data well but could more explicitly state how this study fills a gap in the literature specific to Kayunga, beyond other Ugandan regions like Mbarara or Mulago.

We have improved on the introduction section by providing more information on how this study fills a gap in literature specific to Kayunga. This information has been added for the introduction section. “There remains a critical gap in understanding this burden within semi-rural and peri-urban contexts which present distinct sociodemographic and ecological characteristics including lower caregiver education levels, higher household sizes that may influence nutritional status and anemia risk differently. This study addresses that gap by providing context-specific data on anemia prevalence and its associated factors among HIV-positive children, thereby informing tailored interventions and resource allocation for similar underserved regions.”

Methodology

6. Sampling Justification: Please justify why a sample size of 384 was chosen (e.g., power calculation). Also clarify if this number was based on expected effect size or previous prevalence estimates.

Dear reviewer, we have added the justification of how we reached on the sample size (384). Also this number was based on previous prevalence in a study conducted in Mulago by (Munyagwa, M. (2007).

7. Ethical Statement: It would enhance transparency to mention that written informed consent was obtained from caregivers and assent from older children in the Methods section, not only in the ethics statement.

Dear reviewer, we have added a sub-section of ethical considerations under the methodology section to enhance transparency.

“Ethical considerations

Written informed consent was obtained from the caretakers/guardians of eligible participants aged 7 years and below, while participants aged 8 years and above provided assent by signing an assent-form.

8. Data Collection Tools: Were the structured questionnaires pilot-tested before use? If so, include this information to strengthen methodological rigor.

Under the methodology section, we have added sub-sections of data collection tools, data collection procedures and data analysis. Under the data collection tools, we stated the tools used and where the pretesting was conducted.

9. Mentzer Index Calculation: Define the Mentzer index formula (MCV/RBC ratio) in the Methods section for readers unfamiliar with it.

Dear reviewer, we have added this under the last paragraph of the data analysis section.

“Iron deficiency anemia was assessed using the Mentzer Index, calculated as the ratio of mean corpuscular volume (MCV) to red blood cell count (RBC). Children with a Mentzer Index greater than 13 were classified as having iron deficiency anemia. The proportion of affected children was visually summarized using a pie chart to illustrate the burden of iron deficiency within the anemic subgroup.”

10. Statistical Analysis Software Mention: While SPSS was used for logistic regression, STATA was mentioned later for data analysis. Clarify which software was primarily used and for what parts.

Dear reviewer, we have rectified this. The analysis was done using stata.

Results Section Feedback

11. Table Formatting: Tables are informative but need clearer formatting (e.g., consistent decimal places, alignment, bold headers). Consider using table footnotes where necessary.

We have improved on the tables as advised. The headings of the tables are in bold andjustified the alignment. For the decimal places, for descriptive statistics, we rounded off to 1 decimal place for all the percentages. Bivariate and multivariate analysis results (confidence intervals and p-values), we rounded off to 3 decimal places.

12. Iron Deficiency Anemia Interpretation: The finding that 54.1% of anemic children had iron deficiency may be overestimated if relying solely on Mentzer index. Discuss limitations of using this index instead of serum ferritin or other lab markers.

Dear reviewer, we have added this limitation under study limitations.

“While the study found that 54.1% of anemic children had iron deficiency anemia based on the Mentzer Index (MI >13), this estimate may be inflated due to the inherent limitations of relying solely on this hematological ratio. The Mentzer Index, calculated as MCV divided by RBC count, is a useful screening tool in resource-limited settings but lacks specificity and sensitivity compared to biochemical markers. It cannot distinguish between iron deficiency and other causes of microcytic anemia such as anemia of chronic disease…”

13. Severe Stunting/Wasting Definitions: Clarify how stunting and wasting were classified as "severe" were these based on WHO Z-scores (< -3 SD)? This should be clearly stated.

Dear reviewer, we have clarified on how stunting and wasting were classified. This has been added to the methodology section under data collection procedures.

“Nutritional status was assessed using standard anthropometric measurements, including weight-for-height (WHZ), height-for-age (HAZ), and weight-for-age (WAZ) Z-scores, interpreted according to WHO Child Growth Standards. Stunting, wasting, and underweight were classified as follows:

Moderate stunting/wasting/underweight: Z-score between –2 and –3 standard deviations (SD)

Severe stunting/wasting/underweight: Z-score less than –3 SD

Children with concurrent stunting and wasting (WaSt) were identified when both HAZ and WHZ were below –2 SD. These classifications enabled consistent identification of undernutrition severity and its association with anemia.”

14. Binary Logistic Regression Threshold: The authors used a p-value ≤ 0.2 in bivariable analysis as a cutoff for inclusion in multivariable analysis. Is there a reference supporting this threshold?

Dear reviewer, yes we have added supporting reference as advised. This approach is supported by methodological literature, including Hosmer and Lemeshow’sApplied Logistic Regression, which recommends a relaxed threshold (e.g., p ≤ 0.20 or 0.25) during variable selection to enhance model stability and interpretability.

15. Multivariable Model Fit: Was the model’s goodness-of-fit tested (e.g., Hosmer-Lemeshow test)? Include this to validate the logistic regression results.

Dear reviewer, this has been added under data analysis section of the methodology.

“To assess the adequacy of the final model, the Hosmer-Lemeshow goodness-of-fit test was performed to evaluate whether the observed event rates matched expected event rates across deciles of predicted risk. A non-significant p-value (p > 0.05) indicated that the model fit the data well, suggesting that the logistic regression assumptions were met and the model was appropriate for identifying independent predictors of anemia.”

Discussion and Conclusion

16. Comparison with Previous Studies: The discussion compares findings with studies in Ethiopia, Nigeria, and India, but less attention is given to local Ugandan contexts. Expand on how findings differ or align with national-level policies or programs.

Dear reviewer, we have improved on the discussion section by adding comparisons with some Ugandan based studies and expanded on how the study findings differ or align with national-level policies or programs.

17. Limitations Generalizability: The limitation regarding generalizability due to facility-based sampling is noted. Consider discussing potential selection bias or implications for broader Ugandan populations.

Dear reviewer, we have improved on the study limitations and added the potential selection bias given the fact that the study was conducted in a health facility setting leaving out those not enrolled on ART.

“. Secondly, this study employed a facility-based sampling approach, enrolling HIV-positive children attending the ART clinic at Kayunga Regional Referral Hospital. While this method ensured access to clinical records and standardized laboratory procedures, it may limit the generalizability of findings to the broader pediatric HIV population in Uganda. Children who regularly attend ART clinics may differ systematically from those in the community who are lost to follow-up, not yet enrolled in care, or receiving services from lower-level health facilities. These groups may have differing nutritional profiles, socioeconomic conditions, or disease severity, potentially introducing selection bias. As a result, the prevalence and associated factors of anemia reported here may not fully reflect the burden among HIV-positive children in remote or underserved settings.”

18. Confounding Variables: Did the authors consider adjusting for variables such as dietary intake, micronutrient supplementation, or malaria status in the final model? These could confound the observed associations.

Dear reviewer, these variables were considered and adjusted during the analysis process. This critical information has been added to the data analysis section of the methods.

19. Public Health Implications:The conclusion mentions routine screening and caregiver education. Expand on feasible interventions tailored to resource-limited settings like Kayunga (e.g., community health worker involvement).

Dear reviewer, we have improved on the conclusion section and incorporated some feasible interventions tailored to the study area. This has been added.

“Routine screening for anemia should be integrated into existing ART clinic workflows, using point-of-care hemoglobin testing to enable early detection and management. In addition, leveraging community health workers (CHWs) offers a feasible and scalable strategy for caregiver education and follow-up. CHWs, already embedded within Uganda’s decentralized health system, can be trained to deliver targeted messages on nutrition, iron-rich diets, and adherence to ART and supplementation regimens. They can also facilitate referrals for children with suspected anemia and monitor treatment compliance during home visits. School-based health programs and local radio campaigns may further reinforce awareness, especially in communities with low literacy levels. Strengthening supply chains for iron supplements and multivitamins, alongside capacity-building for frontline health workers, would enhance the sustainability of these interventions. By aligning with Uganda’s Health Sector Development Plan (HSDP) and Community Health Extension Worker Strategy, these approaches offer practical pathways to reduce anemia burden and improve child health outcomes in underserved regions.”

20. Future Research Directions: Recommend future longitudinal studies to explore causal relationships between HIV progression and anemia. Also, suggest investigating the impact of ART regimens on hematologic parameters.

Dear reviewer, we have added a recommendation under the conclusion for longitudinal studies to explore the causal relationship.

“Future research should prioritize longitudinal cohort studies to explore the causal relationships between HIV disease progression and anemia in pediatric populations.”

Editor’s comments: Responses

Minor Editorial Suggestions

Ensure consistency in terminology (e.g., “HIV-positive children” vs. “children living with HIV”).

Dear reviewer, we have ensured consistency in terminology by using only HIV-positive children.

Check all acronyms for first-time definitions (e.g., MI = Mentzer Index).

Dear reviewer, this has been improved as advised.

Avoid repetitive phrases like “in this study” throughout the Discussion.

Dear reviewer, this has been improved as advised.

Use active voice where possible for better readability.

Dear reviewer, this has been improved.

Review figures for resolution and clarity (especially Figure 1 in PDF format).

Figure 1 has been improved as advised.

Dear editor, we have looked through the PLOS ONE guidelines and improved the paper as the journal’s’ requirements.

We note that your Data Availability Statement is currently as follows: All relevant data are within the manuscript and inSupporting Information files.

Please confirm at this time whether or not your submission contains all raw data required to replicate the results of yourstudy. Authors must share the “minimal data set” for their submission. PLOS defines the minimal data set to consist of thedata required to replicate all study findings reported in the article, as well as related metadata and methods.

Dear editor, we have revised our statement on data availability to “Data will be fully available without restriction.” We shall submit all the raw data used for analysis to the journal.

Your ethics statement should only appear in the Methods section of your manuscript. If your ethics statement is writtenin any section besides the Methods, please move it to the Methods section and delete it from any other section. Pleaseensure that your ethics statement is included in your manuscript, as the ethics statement entered into the onlinesubmission form will not be published alongside your manuscript.

Dear editor, we have moved the ethics statement from the declaration to the methods section as advised.

Please upload a copy of Figure 3, to which you refer in your text on page 18. If the figure is no longer to be included as part of the submission please remove all reference to it within the text.

Dear editor, we have removed the figure as advised. Thank you

Reviewer two comments: responses.

The title in the manuscript draft ("Prevalence and Factors Associated With Anemia among HIV-Positive Children in Central Uganda: A Cross-sectional study") differs slightly from the title in the cover letter ("Prevalence and Factors Associated with Anemia among HIV-Positive Children at Kayunga Regional Referral Hospital, Uganda"). Ensure consistency throughout the document.

The title has been slightly changed as advised by reviewer 1 to:

“Prevalence and Determinants of Anemia among HIV-Positive Children in Central Uganda: A Cross-Sectional Study.”

The abstract mentions "Mentzer index >13" for diagnosing iron deficiency anemia (IDA), but the Results section states "MI equal to or more than 13." Clarify the exact cutoff used (e.g., ">13" or "≥13") for consistency.

Dear reviewer, thank you for the clarification, Mentzer index >13 was the used cutoff. This has been improved.

The sample size calculation cites a prevalence of 50.7% from a stu

---

## [Editor Report · Decision Letter 2]

18 Sep 2025

Prevalence and Determinants of  Anemia among HIV-Positive Children in Central Uganda: A Cross-sectional study

PONE-D-25-17782R2

Dear Dr. Herow,

We’re pleased to inform you that your manuscript has been judged scientifically suitable for publication and will be formally accepted for publication once it meets all outstanding technical requirements.

Kind regards,

Alqeer Aliyo Ali, MSc

Academic Editor

PLOS ONE
---

## [Editor Report · Acceptance letter]

PONE-D-25-17782R2

PLOS ONE

Dear Dr. Herow,

I'm pleased to inform you that your manuscript has been deemed suitable for publication in PLOS ONE. Congratulations! Your manuscript is now being handed over to our production team.

Kind regards,

on behalf of

Mr. Alqeer Aliyo Ali

Academic Editor

PLOS ONE